# Assessing the influence of French vaccine critics during the two first years of the COVID-19 pandemic

**Mauro Faccin**[1]*, **Floriana Gargiulo**[2], **Laëtitia Atlani-Duault**[1,3,4,5], **Jeremy K. Ward**[6]

**1** IRD and CEPED, Université de Paris, Paris, France, **2** GEMASS, CNRS, Paris, France, **3** Institut COVID-19 Add Memoriam, Université de Paris, Paris, France, **4** WHO Collaborative Center for Research on Health and Humanitarian Policies and Practices, Université de Paris, Paris, France, **5** Mailman School of Public Health, Columbia University, New York, NY, United States of America, **6** CERMES3, INSERM, CNRS, EHESS, Université de Paris, Villejuif, France

* mauro.fccn@gmail.com

**Data Availability Statement:** The full sets of vaccine-critical URLs, media URLs and tweet IDs can be found on the public repository https://github.com/maurofaccin/datacovvac, along with a set of python scripts for later analysis. The corpus

## Abstract

When the threat of COVID-19 became widely acknowledged, many hoped that this pandemic would squash "the anti-vaccine movement". However, when vaccines started arriving in rich countries at the end of 2020, it appeared that vaccine hesitancy might be an issue even in the context of this major pandemic. Does it mean that the mobilization of vaccine-critical activists on social media is one of the main causes of this reticence to vaccinate against COVID-19? In this paper, we wish to contribute to current work on vaccine hesitancy during the COVID-19 pandemic by looking at one of the many mechanisms which can cause reticence towards vaccines: the capacity of vaccine-critical activists to influence a wider public on social media. We analyze the evolution of debates over the COVID-19 vaccine on the French Twittosphere, during two first years of the pandemic, with a particular attention to the spreading capacity of vaccine-critical websites. We address two main questions: 1) Did vaccine-critical contents gain ground during this period? 2) Who were the main actors in the diffusion of these contents? While debates over vaccines experienced a tremendous surge during this period, the share of vaccine-critical contents in these debates remains stable except for a limited number of short periods associated with specific events. Secondly, analyzing the community structure of the re-tweets hyper-graph, we reconstruct the mesoscale structure of the information flows, identifying and characterizing the major communities of users. We analyze their role in the information ecosystem: the largest right-wing community has a typical echo-chamber behavior collecting all the vaccine-critical tweets from outside and recirculating it inside the community. The smaller left-wing community is less permeable to vaccine-critical contents but, has a large capacity to spread it once adopted.

## Introduction

The COVID-19 pandemic emerged at a tumultuous time for vaccination. The past decade had seen a growing realization among public health and political deciders that reticence towards

of tweets was collected in collaboration with the Science-Po Medialab in Paris using the python scraper Gazouilloire.

**Funding:** This study was funded by Agence Nationale de la Recherche (ANR) through project TRACTRUST, (ANR-20-COVI-0102) Agence nationale de recherches sur le sida - Maladies infectieuses émergentes (ANRS-MIE), project MEDIACAM - ANRSCOV24 to MF and by Agence Nationale de la Recherche (ANR), through project SLAVACO - ANR 20-COV8-0009-01 to JW.

**Competing interests:** The authors have declared that no competing interests exist.

vaccines is widespread [1]. It appears unclear whether doubts are more widespread than in the past because of the lack of longitudinal data in most countries. But many experts have argued that the development of the Internet and online social networks in particular might have allowed doubts towards vaccines to spread by enabling vaccine-critical activists (so-called "antivaxxers") to reach a wider audience [2–6]. Indeed, studies of vaccine-related content posted on social media have shown both that vaccine-critical content is widely available on the Internet and that vaccine-critical activists are very active on social media [6–8]. Early works suggested that vaccine-critical contents tended to be dominant on social media [6]. But studies conducted in the years just before the pandemic tended to show that the scale might be tipping the other way and this could be due to changes in platform-content moderation and monetization rules as well as a growing mobilization of the pro-vaccine community [9–11]. Despite these encouraging news, reticence towards vaccines remains an important issue, to the point that the WHO identified "vaccine hesitancy"—which they define broadly as "the reluctance or refusal to vaccinate despite the availability of vaccines"—as one of the ten biggest threats to global health in 2019 [1].

When the threat of COVID-19 became widely acknowledged, at the beginning of 2020, many tried to see a silver lining and hoped that this pandemic would squash "the antivaccine movement" [12]. How could it have been otherwise? This pandemic was supposed to show what happens when no vaccine is available and therefore convince everyone of the necessity to vaccinate. For months, hopes focused on the development of vaccines against this new disease. However, when vaccines started arriving in rich countries at the end of 2020, it appeared that vaccine hesitancy might be an issue even in the context of this major pandemic. Indeed, the share of the population who intended to vaccinate against COVID was under 70% in several countries including Japan and the USA [13, 14]. In France, only around 45% of the adult population intended to vaccinate against covid when the campaign started just after Christmas 2020 [13, 15]. While vaccine hesitancy has not been a major obstacle in the initial phases of the vaccination campaign, its effect were eventually manifest in plateauing vaccine coverage in some countries including Israel, the USA and France [15–17]. Plateauing vaccine coverage has lead many countries to implement coercive measures such as various forms of health passports and even, more recently, mandatory vaccination [15, 18].

The pandemic does not seem to have squashed "the antivaccine movement" or vaccine hesitancy. But does it mean that the mobilization of vaccine-critical activists on social media is one of the main causes of this reticence to vaccinate against COVID-19? The question of causes is a complex one when it comes to attitudes towards vaccines. They are volatile, complex and highly context-dependent [19, 20] and determined by a great diversity of intertwined factors ranging from the severity of the disease, to the efficiency of the vaccines, to trust in healthcare workers, political deciders and in science very broadly defined [19, 21, 22]. The mobilization of vaccine-critical activists is another factor that meshes with the previous ones as these activists draw on distrust in public deciders and on discourses downplaying the pandemic to convince the largest possible public not to take this particular vaccine and join their cause. They can therefore play a crucial role in translating distrust in institutions and politicians or feelings that the virus is not so dangerous into rejection of the vaccine. A pandemic is a challenge for vaccine critics but also an opportunity. It is an opportunity because health-related issues, and vaccination in particular, become main and heated debates are likely to arise as well as dissatisfaction towards public health and political deciders. Vaccine critics can draw on this to increase their influence.

In this paper, we wish to contribute to current work on vaccine hesitancy during the COVID-19 pandemic by looking at one of the many mechanisms which can cause reticence towards vaccines. We investigate the evolution of vaccine-critical activists' capacity to

influence a wider public on social media. Our work focuses on contents produced on the French-speaking segment of the social media Twitter between March 2020 and October 2021. We draw on a cartography of the main French and francophone vaccine-critical actors and their websites conducted before the epidemic [23] which we updated during the epidemic. We map the evolution of tweets linking to the blogs and websites of these main vaccine-critical actors and compare it to the evolution of tweets linking to the mainstream media as well as the overall vaccine-related contents published in French on Twitter.

Finally, we show that, despite the high activity of vaccine critics, their place in discussions on vaccines has remained relatively constant across the period and very limited compared to mainstream media. Some events allowed them to reach a wider public, including the intense public debate over the efficiency of Hydroxychloroquine as a treatment for COVID-19 that arose in March 2020 and the release in November 2020 of the vaccine-critical documentary "Hold-Up". But overall, their sphere of influence has mainly been restricted to two communities. The largest one composed of far-right conspiracy theorists and was rather closed on itself. The other, much smaller in term of number of users, composed of far-left actors and was somewhat more capable of transmitting vaccine-critical contents to a wider public.

## Data and methods

### Data

Our study combines three independent tweet collections, obtained through a combination of the streaming and search APIs (data collection is performed in real time through the streaming APIs and backward, every week, through the search APIs using [24]). The first dataset, Data-Vac, was collected on the basis of a query focused on vaccine related words, since April 2016. The second dataset, DataCov, contains tweets concerning COVID-19, collected since 2020-03-01. The third dataset, DataHC, contains tweets including keywords that regards Hydroxy-chloroquine and its collection starts on 2020-10-22. Keywords were selected based on the results of an extensive analysis of vaccine-related controversies and vaccine hesitancy in France [9, 23, 25], the full list of keywords can be found online at [26]. All datasets only contain tweets in French.

We merged the datasets filtering each of them with the combined set of query keywords of DataVac and DataCov. After deduplication, our dataset contained 3M tweets, 10M retweets and 840k users. We named this dataset DataCovVac. This is the large-scale dataset on which we build the retweet network and the mesoscale community structure as described hereafter.

In this study we analyze the information flow regarding vaccines and its sources, for this reason we further filtered the tweets to the subset of posts containing URLs pointing to external websites or blogs. We started from two lists of URLs: the first from [23] made up of 285 manually coded URLs of websites and blos of prominent actors that contain vaccine-critical postures, the second comprising 50 URLs of French news media selected by Olivier Duprez as the ones with more readers, and published at https://ymobactus.miaouw.net/labo/. To further extend those lists we searched our database DataCovVac for those URLs that appear in similar scenarios. Starting from the co-occurrence network of all URLs, where two URLs connect to each other if they are shared by the same user, we selected the dilation of the initial unified set of 335 URLs to their nearest neighbors. We visited and manually classified all the URLs of this set, keeping those falling in either of the two categories: vaccine-critical information (382 URLs) and news media (383 URLs). The resulting dataset, with the coded URLs, comprises 209940 tweets containing vaccine-critical URLs (DataCritical) and 1171511 tweets containing media URLs (DataMedia).

## Methods

**Measuring the engagement dynamics.** In order to assess the temporal evolution of users' engagement on Twitter, we take inspiration from compartmental models. At any time each user can be in any of two states:

**engaged (E)** the user has tweeted or retweeted a link to a vaccine-critical website in the last $\tau$ days;

**uncommitted (U)** otherwise.

In this model (inspired by the SIS model in epidemiology [27] where engaged users play the role of the infectious individuals), users become engaged proportionally to their exposure to already engaged peers. The increase of engaged users is proportional to the possible interactions between engaged and uncommitted:

$$dE_t^+ = \alpha_t \frac{E_t(N_t - E_t)}{N_t} \tag{1}$$

where $E_t$ is the number of engaged users at time $t$, $N_t$ is the (possibly time dependent) total number of active users (users that tweeted or retweeted in the last $\tau$ days) and $\alpha_t$ represents the (time dependent) rate at which engaged and uncommitted users come into contact, and the transmission of the information happens.

Analogously, the number of engaged users decreases with time as they become bored with the discussion or change their mind on the issue:

$$dE_t^- = \beta_t E_t \tag{2}$$

where $\beta_t$ represents the rate at which an engaged user looses interest (on average).

The global dynamical system obeys the differential equation:

$$dE_t = dE_t^+ - dE_t^- . \tag{3}$$

Since we can compute directly from the dataset the quantities $dE_t^+$ of users that were uncommitted at $t-1$ but shared a link to a vaccine-critical website at $t$, and $dE_t^-$ of users that were engaged at $t-1$ but did not shared a link within a time window $(t-\tau, t]$, we can estimate the engagement and disengagement rates $\alpha_t$ and $\beta_t$.

The ratio between $\alpha_t$ and $\beta_t$ represents the rate at which the infection spreads over the population and, in epidemiology, is sometimes referred to as *reproduction number*:

$$R_t = \frac{\alpha_t}{\beta_t} . \tag{4}$$

The above can be calculated for the news media URLs as well.

In the present work we select an engagement window of $\tau = 3$ days, although results are robust for slight variations of such parameter. We selected a short time window to better capture the fast-paced changes in a social network as Twitter.

**Directed hyper-graphs.** On Twitter the discussion is usually triggered by one user posting some content and an avalanche of other users retweeting such information helping its diffusion on the social network. The (possibly weighted) retweet network describes how often two users come into contact through the action of retweeting each other. While this model may grasp the structure of user interactions, it fails to distinguish different cascade structures such as users with a high number of singly retweeted tweets or few highly retweeted tweets. Further, the directed nature of the retweeting process is not reflected into the retweet network [28].

To model such flow we choose the directed hyper-graph as a tool that can leverage the interaction between one user (the original poster) and their audience (the retweeting users) [28]. Carletti et al. [29, 30] use a symmetric (non-directed) version in similar settings.

A directed hyper-graph is defined by the pair $\mathcal{H} = \langle V, E \rangle$ where $V$ is the set of nodes and $E$ is the set of directed hyper-edges. Each hyper-edge $e_\alpha = \langle T, H \rangle$ is defined by two paired sets of nodes: $T \subseteq V$ is the tail or source of the hyper-edge and $H \subseteq V$ is the head or sink of the hyper-edge. In our framework the source of information (tail) is the user that posts a tweet, while the head of the hyper-edge is the set of retweeting users.

**Dynamics and hyper-graphs.**   Consider a dynamical system evolving on top of the hyper-graph, in particular a random walker. On a directed hyper-graph the random walk can be defined as follows:

- the walker resides on a node;

- the walker chooses one of the hyper-edges incident to that node on their tail with equal probability;

- the walker crosses the hyper-edge and reaches one of its head nodes, with equal probability.

The above dynamics defines a transition pattern between nodes and allows us to write an effective transition matrix:

$$T_{i \to j} = \frac{\sum_{\alpha \in E(i,j)} |\alpha|^{-1}}{\sum_{k, \alpha \in E(i,k)} |\alpha|^{-1}} \tag{5}$$

where $E(i, j)$ is the set of hyper-edges with tail on $i$ and head on $j$, and $|\alpha|$ is the head size of the hyper-edge $e_\alpha$. With an iterative approach we compute the probability $p(i, j)$ of moving from node $j$ to node $i$ at the steady state. A random walk on a graph, governed by the same effective transition matrix, will precisely reproduce the dynamics on the hyper-graph as defined above.

**Community detection.**   To detect the community structure of the Twitter user base we leverage a dynamical approach such the stability algorithm [31], extended to symmetric hyper-graphs in [29]. The mentioned algorithm maximizes the auto-covariance of the dynamics, projected to the community structure. Since we can compute the transition matrix from Eq 5, and we expect similar community structure from graph and hyper-graph with the same transition matrix, we applied the stability algorithm to Eq 5. We use a unitary resolution parameter which, for symmetric pair-wise graphs, reduces to modularity maximization [31].

In this framework, for any given community $C$, we define the *average visiting probability* as:

$$\frac{p(C)}{|C|} = \frac{1}{|C|} \sum_{i \in C} p(i) \tag{6}$$

where $|C|$ is the number of users belonging to community $C$ and $p(i) = \sum_j p(i, j)$ is the steady state distribution of the random walk dynamics. We further define the *escape probability* as:

$$p(\bar{C}|C) = \frac{\sum_{i \notin C, j \in C} p(i, j)}{\sum_{k \in C} p(k)} \tag{7}$$

where $\bar{C}$ is the set of users not in $C$. The former describes the participation of the community nodes in the random walk dynamics, while the latter represents the tendency of the random walker to leave the community.

## Results

### How much does vaccine-critical information circulate on Twitter?

The first question that we addressed is whether the circulation of vaccine-critical information on Twitter increased during the pandemic. To do so, we first globally analyzed the number of tweets concerning the whole vaccine debate in France (from DataVac), in the period between March 2020 and October 2021, and we compared this with the tweets containing a link to a vaccine-critical URL (DataCritical) and with the tweets containing a link to a media URL (DataMedia).

In the lower plot of Fig 1 we observe that the volume of tweets on vaccines is not constant during the period: the volume had a first striking increase at the beginning of November 2020 and a second one at the beginning of June 2021. We can therefore identify three periods with different levels of engagements of the user base with the vaccine topic.

The first volume growth is associated to a significant event related to the pandemic: on the 8th of November 2020 Pfizer announced that their vaccine finished the trials phase and was ready to be distributed. The availability of COVID-19 vaccines shifted the debate from an abstract discussion about potential vaccines to the concrete focus on the actual vaccination campaign.

Notice from Fig 2 that the first period, before the 11th of November 2020, is focused on generic hashtags either relating to the virus and the government's handling of this pandemic (#virus, #veran, #chloroquine, #oms, #raoult, #masques) or relating to vaccines (#billgates, #bigpharma, #trump). On the contrary, the second period is characterized by the presence of hashtags mostly related to the vaccines and the companies behind their development.

In the second period, the average volume of tweets settles on a stable quantity that is around 7 times the volume observed before. We can also notice that this general increase is also visible in the media (DataMedia) and in the vaccine-critical URLs datasets (DataCritical).

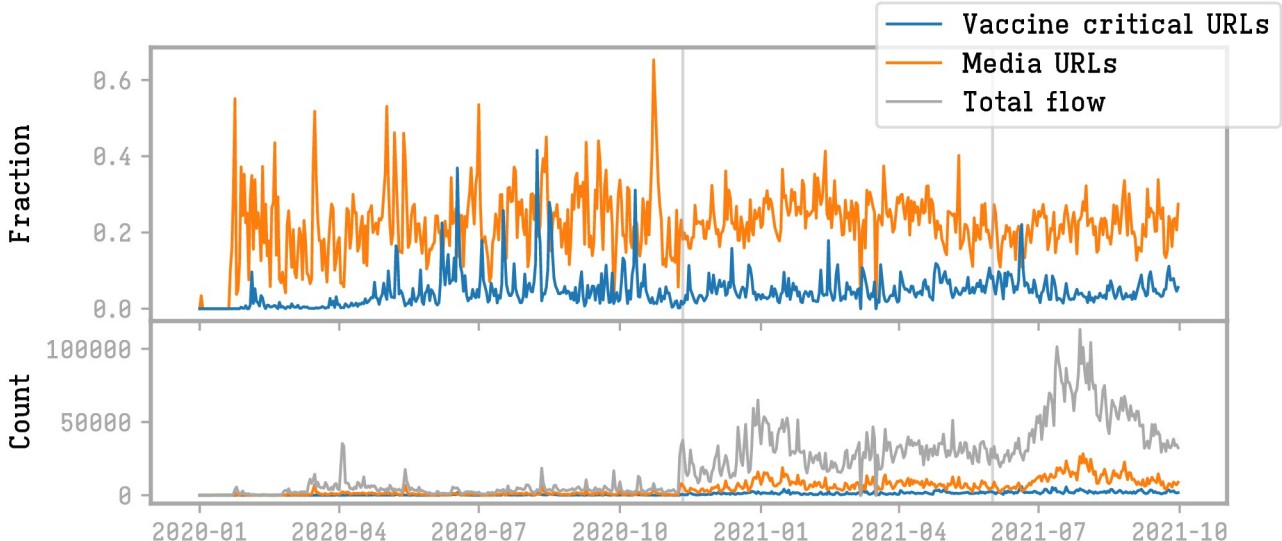

**Fig 1.** Upper plot: Daily fraction of tweets and retweets containing a media URL (orange) or a vaccine-critical URL (blue). ower plot: number of tweets and retweets in the whole dataset (gray), from media (orange) and vaccine-critical URLs (blue).

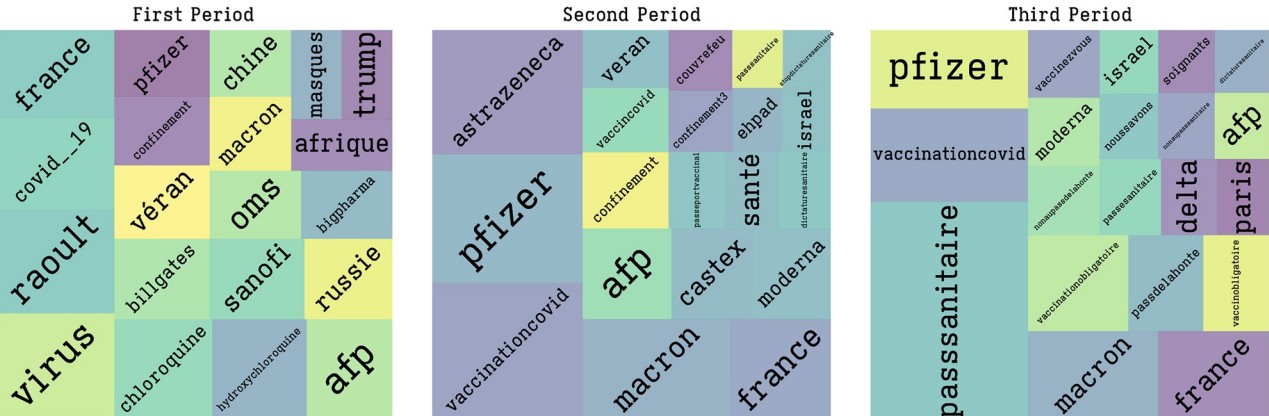

**Fig 2. Treemap of the most important hashtags per period.** Pre-Pfizer announcement; post-Pfizer announcement; Health pass discussion.

The third period, characterized by an even higher volume of tweets, starts at the beginning of June 2021. Discussion focuses on the health pass (#passsanitaire), Fig 2. In this case we observe a constant increase of the volume until the end of August 2021 followed by a decrease to the initial situation. At the peak, the volume increased to 28 times the average of the second period.

Going back to our initial question on the possible increase of the circulation of vaccine-critical contents, we can observe from the upper plot of Fig 1 that, independently of the periods, the fraction of tweets containing media URLs and vaccine-critical URLs remained almost constant during the entire observation window. This means that the overall relative space occupied by the information from media and vaccine-critical sources remains constant (upper plot of Fig 1). In particular, vaccine-critical content did not gain ground.

We also addressed the question of the possible increase of vaccine-critical presence in the debate, from the point of view of the users, using the engagement metric and the $R_t$ estimation described in the methods' section.

Users that tweet or retweet a piece of information show their engagement with the content of that information. In particular tweets sharing links to well known web pages with vaccine-critical content, show that the original poster and, probably to a lower degree, all the subsequent retweeters endorse that content.

We analyze the dynamics of the users' engagement in spreading the information from the vaccine-critical ecosystem, namely the individual propensity to tweet or retweet posts containing a vaccine-critical URL.

Users may endorse vaccine-critical online contents via a constant tweet or retweet flow or with sporadic sharing of information. We define a user as engaged if in the past $\tau = 3$ days tweeted or retweeted a link to any such online content (the following results are robust to the change of the time frame $\tau$). We measure the number of users engaged in the spreading of vaccine-critical information and the reproduction rate $R_t$ of vaccine-critical information in our portion of the Twittosphere (the number of users that became engaged via contact with the posting of the previously engaged) according to the measures described in the Methods section. We compare these measures to the corresponding measures for the diffusion of news media.

Fig 3 displays the temporal behavior of the two discussed measures: the volume of engaged users and the reproduction rate of vaccine-critical information (blue) and of news media

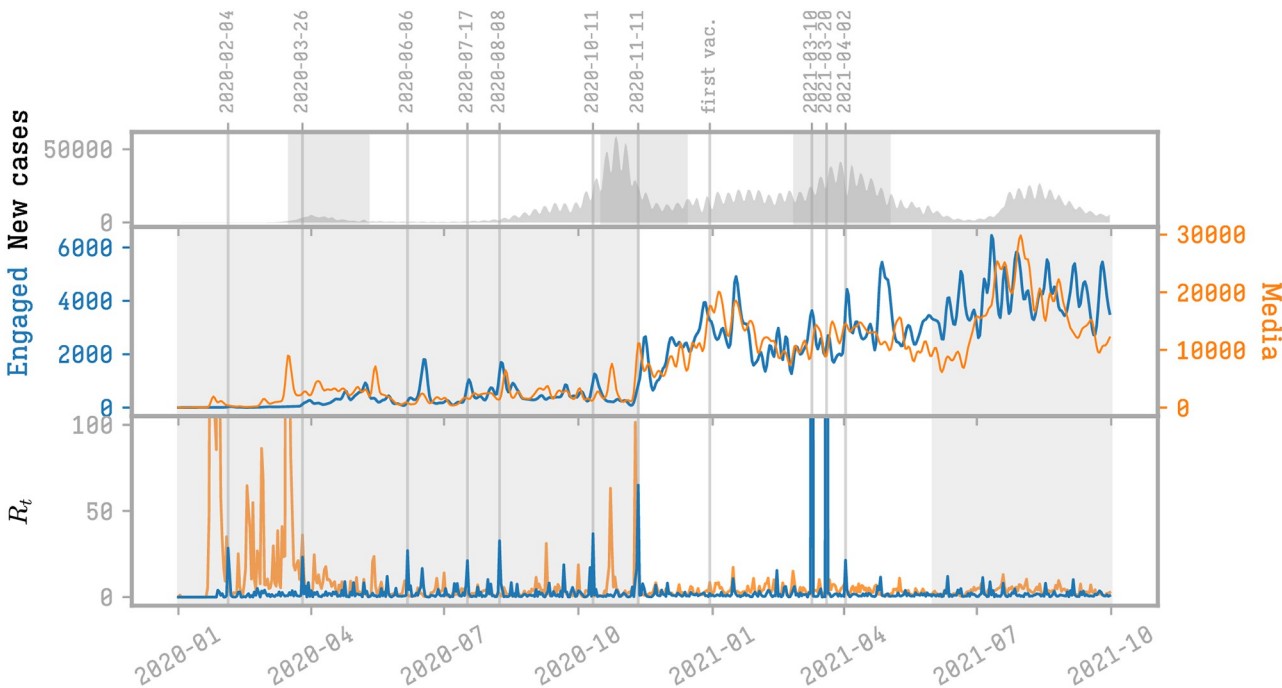

**Fig 3. Engagement of users with vaccine-critical content.** Above: evolution of the total number of users engaging with news media (orange) or with vaccine-critical contents (blue). Below: the reproduction number shows peaks of engagement around some events. The greyed area on the top panel is proportional to the daily new cases in France.

(orange). In the upper plot we also show the pandemic scenario (the number of cases and the lockdown periods) to visually check if the engagement scenario is possibly connected to the pandemic landscape.

The number of engaged users does not strictly follow the volume of the discussion, both for vaccine-critical contents and for the media and in particular we do not observe a striking increase in the third phase. The temporal evolution of the reproduction number shows few events triggering a noticeable response.

The engagement with vaccine-critical information experiences three different growing phases, whose beginning can be identified by relevant peaks of the $R_t$ index. Analyzing the main hashtags emerging around the $R_t$ peaks, we can observe that these phases are not connected to the evolution of the pandemic. The first triggering event, on the 26th of March 2020, falling during the first lockdown and next to the first pandemic peak, is connected to the hashtags #hydroxychloroquine and #raoult and is related to the government announcement of forbidding the use of hydroxychloroquine treatments for COVID-19. The second event, coinciding with the transition to the second period on the 11th of November 2020, connects to the Pfizer vaccine, but also to the release of the vaccine-critical documentary "Hold-Up". The number of users engaged in the diffusion of vaccine-critical contents increases until February 2021, and decreases quickly after this date. A third slower growing phase starts one month later, related to a large triggering event on the 10th of March 2021. This third long-lasting growing phase begins with the controversy on the retraction in Denmark of the Astrazeneca vaccine (hashtags: #astrazeneca, #denmark), but also with the first declarations about a future installation of the European health pass (#passsanitaire). This phase, connected to the debate on the health pass experiences a growing phase until the effective

**Table 1. Events detected.** List of events that triggered a rapid growth of engagement within Twitter users.

| date | topic of the event |
|---:|---|
| 2020-02-04 | COVID-19 comes from a lab experiment. |
| 2020-03-26 | Retraction of hydroxychloroquine treatment. |
| 2020-06-06 | Adverse events and a death cases in COVID-19 critical trials. |
| 2020-07-17 | Hydroxychloroquine and Remdesivir instead of Bill Gates' vaccine |
| 2020-08-08 | "Il faut refuser ces vaccins contre le COVID-19!" Dr Pierre Cave |
| 2020-10-11 | First appearance of #stopdictaturesanitaire |
| 2020-11-11 | Pfizer announcement / holdup movie |
| 2021-03-10 | Suspension of Astrazeneca in Denmark |
| 2021-03-24 | More suspension of Astrazeneca |
| 2021-04-02 | Other reported cases of "death by vaccine" |

installation of the pass in August 2021 and stabilizes thereafter. Other important maxima of the $R_t$ value are reported in Table 1.

Engagement with media content has a more marked tendency to follow the shape of the pandemic cycles, with the exclusion of the asynchronous behavior during the intermediate period and related to the discussion connected to the introduction of the Pfizer vaccine that, together with the increase of the discussion volume, also forced the engagement of new users.

The asynchronicity of the $R_t$ peaks for vaccine-critical and media engagement, with the exclusion of the 11th of November 2021, shows that the vaccine-critical activity does not strictly follow the media agenda that in its turn is more strictly connected to the evolution of the pandemic.

## Which actors spread vaccine-critical contents on Twitter?

In the previous paragraph we analyzed which part of the vaccine debate is occupied by vaccine-critical contents. Now we will enter more deeply inside the structure of the information flows, and we will study which kind of actors are present in the debate and their role in spreading patterns.

**The hyper-graph structure and mesoscopic shape of the debate over COVID-19 vaccines.** The distribution of tweets and retweets containing vaccine-critical URLs can be seen as a proxy of the reach of the vaccine-critical discourse on Twitter. The most commonly used structure to represent the information flows on Twitter is the retweet network, representing the directed links among users who retweeted each other. However this structure can hide a potential bias by not allowing to distinguish two crucially different situations: a user who is retweeted $N$ times for a single tweet and a user whose $N$ tweets are individually retweeted one single time. In both cases, this user would have an out-degree $k_{out} = N$ in the retweet network, but the overall dynamics of the information flows would be fundamentally different.

To overcome this problem we use a higher-order network representation introducing a directed hyper-graph structure, composed by nodes and hyper-edges [30]. In our case each hyper-edge represents a tweet and its retweet cascade, in particular the original poster is the source or tail of the hyper-edge while the retweeting users represent the sink or head of the hyper-edge. From the hyper-retweet graph we analyze the dynamical properties of the graph, as described in the methods section.

We analyze the community structure of Twitter users discussing COVID-19 vaccines through the partition of the hyper-retweet network (from DataCovVac). Such communities represent densely connected groups of users among which information circulates quickly. In

**Table 2. Communities and their respective qualitative interpretation.**

| Community | Interpretation |
|---|---|
| $C_0$ | media aggregators or French web influencers. |
| $C_1$ (and secondly $C_{24}$) | Far right groups. |
| $C_2$ (and $C_{15}$ and $C_{25}$) | public health institutions, medical doctors and associations |
| $C_3$ and $C_4$ | French and international news media. |
| $C_5$ | Far left and trade unions. |
| $C_6$ | government representatives. |
| $C_7$ | other French-speaking countries (Canada) |
| $C_{9,10,11,14,17}$ | other French-speaking countries (Belgium, Morocco, Switzerland,...) |
| $C_{12,16,23,30}$ | non French-speaking countries (India, Israel,...) |
| $C_{8,13,18,19,20,27}$ | local French institutions (Nord-Picardie, Loire, ...) |

particular the information flow over this mesoscopic structure of the network unveils the ability of the vaccine-critical users to reach other users beyond the natural extension of their *social bubble*. The community structure is determined by the information flow through the maximization of covariance. This approach is akin to modularity maximization [32] when a random walk is associated to the graph, in particular we consider the generalization of the Louvain algorithm to directed graphs as in Dugue et al. [33].

The community detection algorithm identified 2991 communities in the largest connected component. However, the 30 larger communities concentrate more than 90% of the users. Let us focus on these larger communities, starting with the analysis of their social composition. Performing a qualitative analysis of the most active user profiles of each community, we can infer a community profile, see Table 2.

To further characterize the content shared by the different communities we calculated their hashtag preference profiles. We cluster communities and hashtags based on the over-usage of hashtag $a$ on community $i$:

$$\xi_{ia} = P(i, a) - P(i)P(a) \qquad (8)$$

where $P(i, a)$ is the frequency of hashtag $a$ in the community $i$, $P(a)$ is the global frequency of hashtag $a$ in all the tweets containing the 50 top hashtags and $P(i)$ is volume of tweets of community $i$ (limited to the 50 largest communities). A negative value means that the hashtag is less used by the community compared to a random situation while a positive value indicates an over-usage of the hashtag. We performed a hierarchical clustering based on the score matrix, and we classified 7 groups of communities according to their hashtag preferences. The clustering is displayed in Fig 4.

The configuration emerging from the clustering shows a clear preference of the right-wing communities ($C_1$ and $C_{24}$) toward the vaccine-critical ecosystem: from #bigpharma, to the keywords related to alternative drugs (#raoult, #invermectine, #hydroxychloroquine,...), to the opposition to the health pass (#nopasssanitaire, #dictaturesanitaire,...). Notice the large use, in these communities, of vaccine producers' names (e.g. #moderna and #pfizer). As we noticed in a previous paper [9], the vaccine-critical galaxy has a marked capacity to capitalize on hashtag use in order to be easily retrieved in keyword searches. The large use of these hashtags by a community implies that a user looking for information on a particular vaccine, has a higher probability to reach contents from this community. The vaccine-critical communities have a low propensity to tweet institutional hashtags (#macron, #gouvernment, etc.) and public health hashtags (#vaccinecovid,

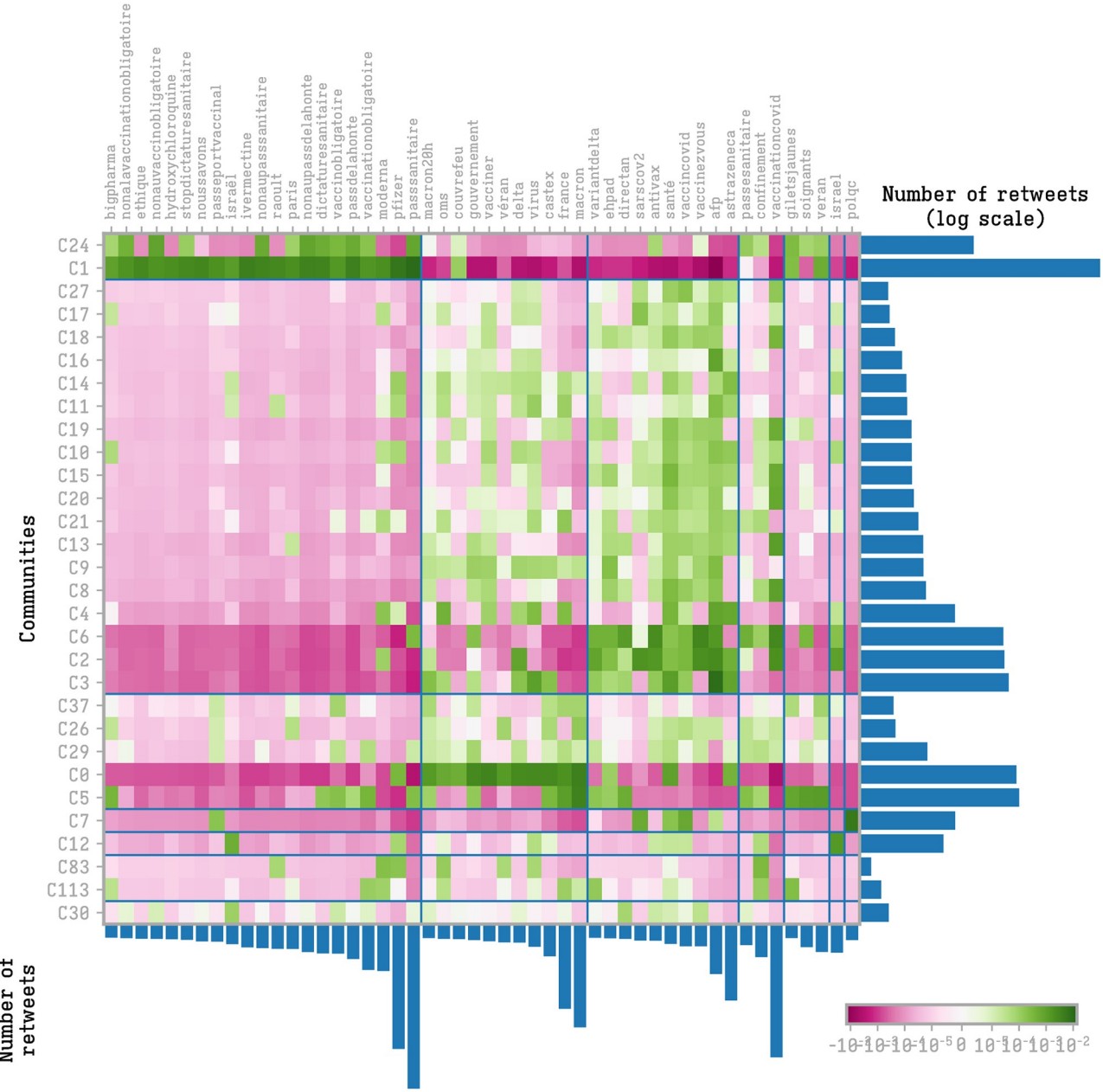

**Fig 4. Hashtags and communities meta-structures.** Hierarchical clustering of the community hashtag preference profiles.

#vaccinezvous, etc.). These communities also have a high tweeting activity concerning the yellow-vests movement.

The set of hashtags concerning government and public health are the most spread by the media, public-health actors and by local and national political institutions. Notice in particular the similar use of public-health connected tags among $C_2$, $C_3$ and $C_6$. The hashtag #antivax is classified in this group, showing that the labeling of the vaccine-critical movement is not the product of a self-definition by the vaccine-critical users, but rather from a media driven reconstruction.

Notice also that the left-wing community $C_5$ shares vaccine-critical hashtags, concerning above all criticism of #bigpharma, but in connection with other tags pointing to politics and social movements.

The general community of web actors, $C_0$, is the one with higher and most exclusive focus on politics.

To better understand the internal structure of the vaccine-critical ecosystem we repeated the previous analysis on use of vaccine-critical URLs. Here we observe a clear bi-partition of these URLs, see Fig 5. The ones connected to the right-wing galaxy, lead by "childrenhealthde-fense.org" and "lemediaen442.fr", that diffuse inside the major right-wing community, $C_1$. On the other side, some URLs, which contains less explicitly "antivaccinationist" and conspiracy theory contents, like "reseauinternational.net" and "media-presse.info", are largely used by the left-wing community and have a large spread among all the other communities.

**The mesoscale structure of the information flow.**   We will now analyze how the information flows between the communities. To estimate this we calculate the probabilities for a random walker to get out of a community (following the retweet hyper-graph structure) and the probability to visit a node of a given community (average visit probability), as defined in Eqs 6 and 7 respectively. Notice that, in this analogy, the walker can be considered as a piece of information. We distinguish the full hyper-graph (the one in which we compute the community structure), the hyper-graph only including tweets with vaccine-critical URLs and the one including only media URLs. Fig 6 illustrates these measures.

Communities are groups of users with high inner and low outer information flow. This express, on the right plot of Fig 6, on low escape probabilities for all the communities. The average visit probability depends on the (average) activity of the group users, the higher the retweeting activity of a group, the higher will be the probability that a hyper-edge originates or terminates in that community.

This picture changes completely if we just consider vaccine-critical URLs. In this case we observe a behavior polarization in the two largest communities, $C_0$ and $C_1$. Community $C_1$, the largest right wing group, acts more as an echo chamber: most of the tweets its recirculate within itself. However, it acts to a lower degree as a filter bubble, showing a remarkable absorption of vaccine-critical contents produced in other communities. Community $C_0$, on the contrary has a low permeability to vaccine-critical contents, with a low internal circulation and a generally low probability to be reached. The other large communities organize on the diagonal between these extremes.

Considering media URLs, the picture reproduces similar but less heterogeneous results. Community $C_0$ shows its main role in information diffusion having a balanced rate among internal and external retweets, still maintaining a low capability to collect information (low retweeting activity). Community $C_1$ has a lower probability to be reached via media contents, but it has a higher probability of diffusing outside its boundary.

We can also see that the highest polarization in terms of roles in the information ecosystem is not observed among left and right-wing but rather among the right-wing ($C_1$) and some important characters of the French Twittosphere ($C_0$), i.e. the expert users of the online platforms. The left-wing community ($C_5$) has an intermediate position, as in Fig 6. Notice that this central position in the plot makes this community a key actor in the diffusion of vaccine-critical information: even if it remains less receptive to vaccine-critical contents, this community has a higher capacity to spread this kind of information to a larger public. On the other hand, the far-right community has the typical behavior of an echo chamber with only internal diffusion.

We also notice that the escape probability for each community changes in time, in particular in the case of communities in intermediate positions ($C_{2...6}$), see Fig 7. Larger communities

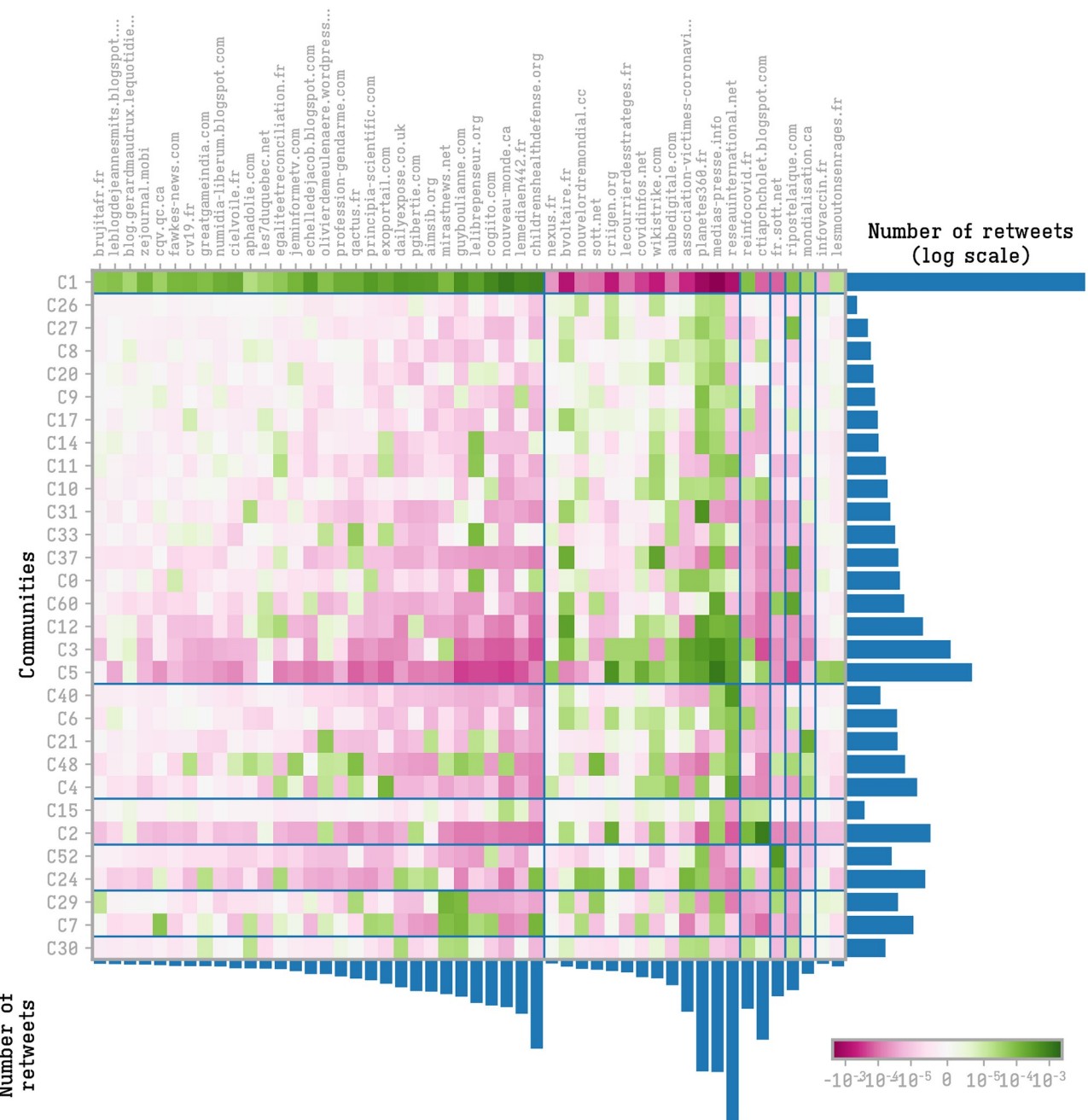

**Fig 5. Vaccine-critical URLs and communities meta-structures.** Hierarchical clustering of communities based on their URLs sharing profile and vice versa.

($C_0$ and $C_1$) keep a stable role throughout the three periods. While $C_2$ and $C_6$ (health and government figures) find the maximum outward reach in the middle period, communities $C_3$ and $C_4$ (French and international media) find their minimum reach in the same period. The left-wing community $C_5$, on the other hand, increases its escape probability in the last period, with discussions about the health pass.

Similar changes are not found while sharing media URLs nor in the full dataset (central and right part of Fig 7).

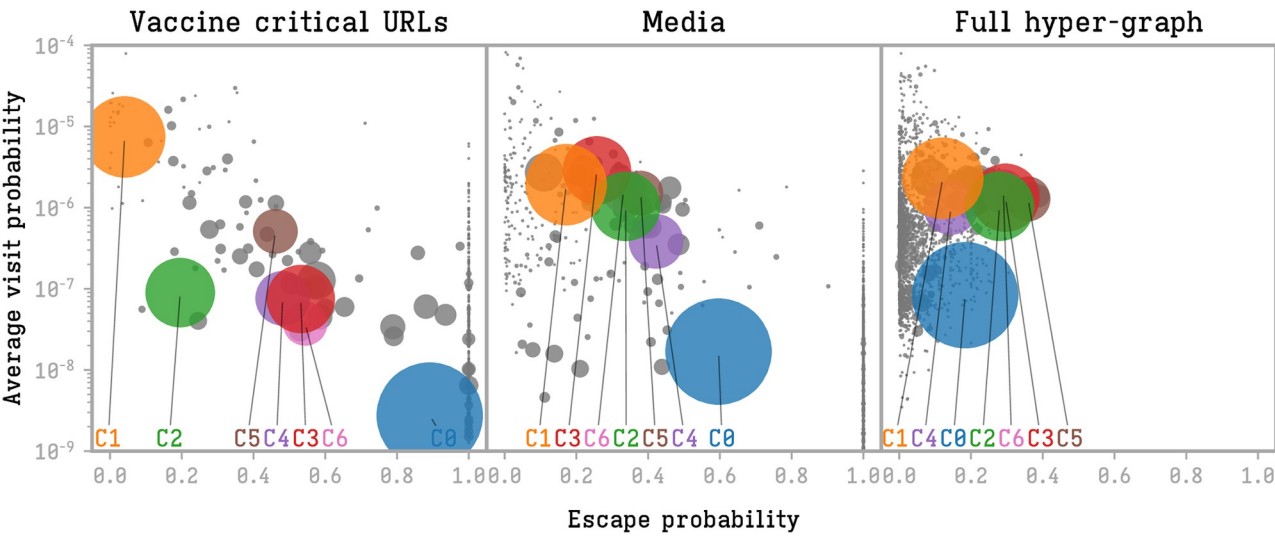

**Fig 6. Escape probability and (average) visit probability of each community on the hyper-graph structure.**

## Discussion

The main outcome of this work is that, in the Twitter ecosystem, the relative reach of the vaccine-critical activists has remained constant and limited in comparison to that of the mainstream media.

The first main implication of our results pertains to current discussions of the spread of misinformation on social media. Our results echo the recent works suggesting that initial assessments of the prevalence of fake news, conspiracy theories, misinformation and disinformation on the Internet might have over-estimated the importance of these phenomena [34–36]. Regarding vaccines, it is also possible that our results illustrate the effects of the changes in algorithms and moderating rules made by platforms to address this issue. Nevertheless, it is

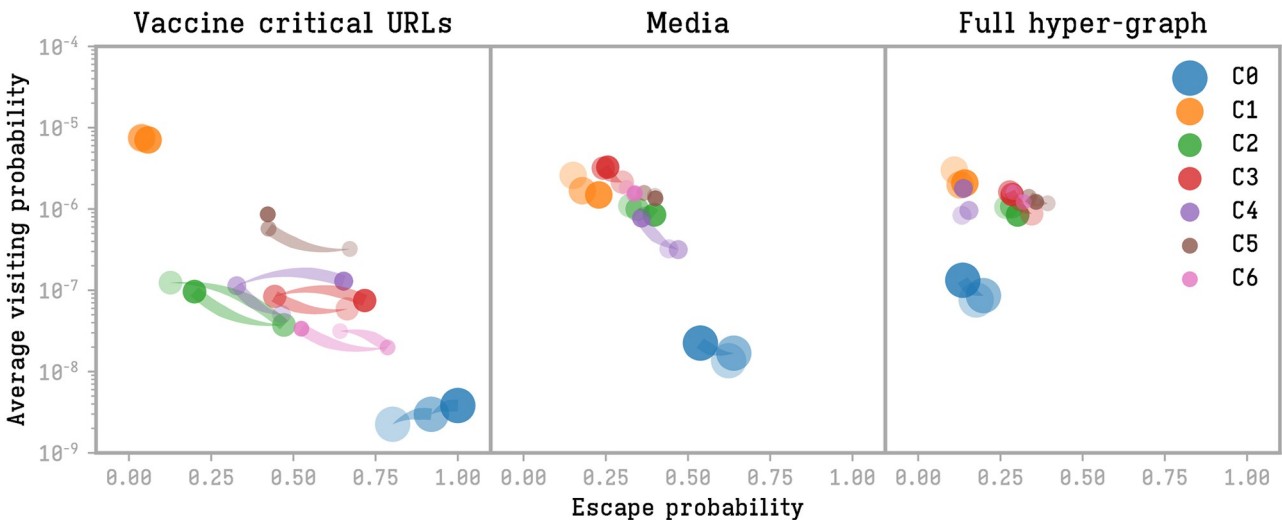

**Fig 7. Escape probability and (average) visit probability changes in the three detected periods.** Community size is proportional to its number of users. Only the larger communities are reported for the sake of the figure readability.

important to note that some events allowed vaccine critics to reach a much wider public than on average during the period, to the point where their reach was comparable to that of the mainstream media. Among these events we would like to comment specifically on the heated controversy over the efficacy of hydroxychloroquine as a treatment against COVID-19. The importance of this event, at the end of March 2020, in our data, points to a crucial element in the understanding of both vaccine hesitancy and vaccine critics' ability to seize opportunities such as the COVID-19 pandemic to reach a wider audience. It should remind us that doubts regarding vaccines and criticism of vaccines are never just about vaccines. They are also about trust in public health authorities, in agencies in charge of authorizing medical products and monitoring their safety, in mainstream scientific research and even about politics [4, 19, 21].

Symmetrically, this means that vaccine critics can seize the opportunity of public debates arising over issues that do not concern vaccination specifically but do engage with these broader issues. The debate over hydroxychloroquine fits this bill perfectly. France was at the center of the international controversy over this specific drug. The debate raged for weeks in the French mainstream media and took on a very politicized turn with proponents of hydroxychloroquine casting doubt on the way clinical research on COVID-19 is performed, the severity of the disease and the probity of researchers, public agencies and the government [37, 38]. Our data suggests that, by pushing forward the types of themes and arguments historically associated with vaccine criticism, this other controversy allowed vaccine critics to attract a wider audience to their own cause by making it part of a broader cause. This echoes the results presented in studies of debates over hydroxychloroquine on social media which tend to show that the community of hydroxychloroquine defenders and that of critics of the COVID-19 vaccines overlap greatly [39]. This result has implications. While these events do not seem to have been associated with a stronger presence in the overall discussions on vaccines on Twitter, it is possible that they do allow vaccine critics to increase their influence on the wider public in little incremental steps. It is possible that these events help them plant the seed of doubt in new audience who won't engage regularly or ever again with vaccine-critical contents but will remember that vaccines are "debated". This is important because the perception that there exists a controversy over a given scientific subject is at the core of most instances where people's beliefs deviate from the scientific consensus [40–42].

The second main implication of our results pertains to the relationship between what we can observe on social media and the evolution of public attitudes towards vaccines in the whole population. We found that vaccine critics' place in overall discussions of vaccines has remained relatively constant across the period. This does not mirror existing data on the French public's attitudes to the COVID-19 vaccines during the pandemic. Intentions to vaccinate against COVID-19 remained constant at around 75% from March 2020 to May 2020, then decreased steadily until the end of December 2020 when they were as low as 45% before they increased relatively steadily to reach around 80% in July 2021 [15, 43]. This mismatch in both trends, combined with the much wider place occupied by mainstream media in the debates over vaccines on Twitter, suggests that the impact of vaccine-critical mobilizations on social media is limited. This finding is consistent with other works showing that mainstream media remain a dominant influence on discussions on social media and on people's perceptions [35, 44, 45]. Many studies have shown that a lot of misinformation is spread widely on social media not by social movements and radical actors, but by mainstream media when they give voice to mainstream actors such as politicians [35, 44, 46]. This echoes older debates over the role of the traditional media during pandemics. During and after the H1N1 flu pandemic of 2009-2010, many wondered whether online social media would replace traditional media as the main sources of information for the public (for a review of the literature, see [47]). While our results do not directly compare the sources of information used by the public, we

nevertheless can see that traditional media remain very influential even on social media. This underlines the responsibility that mainstream media and politicians have in informing the public and in legitimizing fringe or radical points of view [48].

The mismatch between trends on Twitter and data collected via more traditional methods has been underlined in many studies [49]. It points to the over-representation on social media of very active minorities [50, 51]. While it is possible that we would have found less of a mismatch had we turned to a different social media, such as Facebook [52], these results point to the limits of drawing on social media data to address issues pertaining to vaccine hesitancy. But, they also suggest pathways to understanding the reason for this mismatch as well as the reasons why social media have a limited influence. Indeed, we found that vaccine-critical contents were mainly shared by two relatively closed communities, one to the far-right and one to the far-left—the latter being somewhat more able to spread this content beyond their boundaries. Both communities tied vaccination with wider political tropes but also with conspiracy theories. This result points to the shifting French media landscape which seems to be less and less structured along a left-right axis and increasingly oppose institutionalist outlets to anti-elites [45]. The latter is mainly associated with a complex ecology of websites and social media platforms as mainstream media have committed to gatekeeping against many forms of political and scientific critique, including on issues of vaccination [23, 53]. Another reason is that most anti-elite political parties (*Rassemblement National* to the far-right and *France Insoumise* to the far-left) have strived to avoid being too radical in their critique to maintain their chances of gaining power. This has meant that the more radical actors on the far-right and far-left, conspiracy theorists and vaccine-critical activists did not benefit from the visibility offered by the mainstream media and had to over-invest on the Internet as a tool for reaching the public. This points to the importance of the ecology connecting social media, traditional mainstream media and political actors. Indeed, sociological studies of the determinants of mainstream news cycles have shown that journalists and political actors coordinate (involuntarily and informally) to set the boundaries of political debates and of what is defined as too radical [53–55]. This has a crucial impact on the ability of various actors, including vaccine critics, to reach a wide audience and make their arguments appear legitimate. It seems that in the case of COVID-19 vaccine, vaccine criticism has remained somewhat confined to the margins—even though hesitancy was very much common! The main far-right and far-left parties (*Rassemblement National* and *France Insoumise*) have adopted very ambivalent positions. Their representatives very rarely explicitly criticized vaccination in itself but were critical of most aspects of the organization and targets of the vaccination campaign. On the far-right, more marginal actors, such as Florian Philippot and Nicolas Dupont-Aignan, clearly endorsed a vaccine-critical rhetoric. As for the mainstream media, overall they gave little visibility to vaccine critics. However, it is important to note that these are precarious adjustments. Indeed, several mainstream media such as the TV channel CNEWS and the radio Sud-Radio have consistently given voice to vaccine skeptics and covid deniers. Also, at the time of writing of the paper, the leaders of both France Insoumise and the Rassemblement National have adopted very vaccine-critical stances—Marine Le Pen (RN) standing against vaccination of children and Jean-Luc Mélenchon (FI) letting his supporters boo COVID-19 boosters during a meeting. More structurally, a number of media recently acquired by French billionaire Vincent Bolloré have shifted their editorial line to one closer to the far-right and to scientific populism. There is a risk that this type of shift could lead to less scientific gatekeeping, normalization of vaccine criticism and the integration of vaccine skepticism as part of a major political community's identity, as was seen in the USA in the last decade in part due to the influence of Fox News on the political landscape [41, 56].

## Author Contributions

**Conceptualization:** Jeremy K. Ward.

**Data curation:** Mauro Faccin.

**Formal analysis:** Mauro Faccin, Floriana Gargiulo, Jeremy K. Ward.

**Investigation:** Jeremy K. Ward.

**Methodology:** Mauro Faccin, Floriana Gargiulo.

**Project administration:** Laëtitia Atlani-Duault.

**Supervision:** Laëtitia Atlani-Duault.

**Validation:** Floriana Gargiulo.

**Visualization:** Mauro Faccin.

**Writing – original draft:** Mauro Faccin, Floriana Gargiulo, Jeremy K. Ward.

**Writing – review & editing:** Mauro Faccin, Floriana Gargiulo, Laëtitia Atlani-Duault, Jeremy K. Ward.

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
