## [Decision Letter · Decision Letter 0]

13 May 2022

PONE-D-22-05617Assessing the influence of French vaccine critics during the two first years of the COVID-19 pandemicPLOS ONE

Dear Dr. Faccin,

Thank you for submitting your manuscript to PLOS ONE. After careful consideration, we feel that it has merit but does not fully meet PLOS ONE’s publication criteria as it currently stands. Therefore, we invite you to submit a revised version of the manuscript that addresses the points raised during the review process.

Please note that all three reviews recognize the merit and significance of your work -- but they also identify a number of minor weaknesses. In my opinion, these weaknesses can be sufficiently addressed as long as you thoroughly follow the suggestions, or respond to the concerns, of the reviewers.

We look forward to receiving your revised manuscript.

Kind regards,

Constantine Dovrolis

Academic Editor

PLOS ONE

Journal Requirements:

2. In your Methods section, please include additional information about your dataset and ensure that you have included a statement specifying whether the collection and analysis method complied with the terms and conditions for the source of the data.

“This research has benefited from the financial support of the Agence Nationale de la Recherche (projects TRACTRUST - ANR-20-COVI-0102 and SLAVACO - ANR 20-COV8-0009-01) and the ANRS-MIE (project MEDIACAM - ANRSCOV24)”

Reviewers' comments:

Reviewer's Responses to Questions

**Comments to the Author**

1. Is the manuscript technically sound, and do the data support the conclusions?

Reviewer #1: Yes

Reviewer #2: Partly

Reviewer #3: Partly

2. Has the statistical analysis been performed appropriately and rigorously? 

Reviewer #1: Yes

Reviewer #2: N/A

Reviewer #3: Yes

3. Have the authors made all data underlying the findings in their manuscript fully available?

Reviewer #1: No

Reviewer #2: No

Reviewer #3: Yes

4. Is the manuscript presented in an intelligible fashion and written in standard English?

Reviewer #1: Yes

Reviewer #2: Yes

Reviewer #3: Yes

5. Review Comments to the Author

Reviewer #1: Summary:

In this paper, the authors analyze the debates over the COVID-19 vaccine on the French-speaking segment of Twitter. First, they study the evolution of vaccine-related debates. They show that specific events increase the reach of the vaccine-critical activists on social media, but this information flow is relatively limited compared to mainstream media. Second, they analyze the community structure of discussions and examine how information flow between communities. Authors conclude that the largest far-right community is the echo chamber of conspiracy theorists. In contrast, a smaller community that consists of far-left actors is more capable of communicating vaccine-critical content to a broader public.

They assess the evolution of user engagement using a model similar to the SIS network epidemic model. Here, users that share vaccine-critical information are analogous to infectious individuals in the SIS model. Furthermore, their community analysis relies on hypergraphs derived from retweet data. Hypergraph helps differentiate a user who is retweeted N times for a single tweet and a user whose N tweets are individually retweeted one single time, which is crucial for capturing the dynamics of the information flow.

In conclusion, I think this paper presents a rigorous analysis of vaccine hesitancy on Twitter during the COVID-19 pandemic. I think the methodology is accurate, and the results are significant.

Minor Issues:

Some points regarding data collection are unclear to me:

(1) To the best of my understanding, "vaccine-critical URLs" refer to websites other than mainstream media. For example, websites of prominent actors in vaccine controversies. It may be helpful for the reader if this is reminded in the data subsection.

(2) "we searched our database for those URLs" is the database refer to dataset DataCovVac?

(3) I could not fully follow how the co-occurrence network is used to label URLs automatically? Did you also propagate the labels of media URLs to closest neighbors? It appears it is performed only for 285 vaccine-critical URLs. If this is the case, it is not clear how 382 media URLs were obtained from the initial 50 URLs.

I think that a figure (e.g., a flowchart) that explains the data collection process might be helpful. However, this is not a necessity.

I think the authors could mention the motivation behind using hypergraph instead of a standard retweet network earlier in the method section. I think it is a crucial choice, and the reason is explained in the middle of the results section.

Sometimes the COVID-19 outbreak is mentioned as an epidemic, while it is sometimes referred to as a pandemic. Is there a nuance based on the context, or are the words "epidemic" and "pandemic" used interchangeably? For example, the title says "COVID-19 pandemic" and the abstract says "COVID-19 epidemic" this might confuse the reader.

Reviewer #2: In this study, the authors investigated the influences and spreading of vaccine-critical content on social media. They focused on Twitter data and applied network analysis tools to answer two questions: (1) Did vaccine-critical contents exhibit a "rise" during the COVID breakout? (2) What are the roles that different communities (groups of

closely-connected Twitter users) play in the flow of vaccine-critical information.

Generally, the draft is clear about the questions and the general approaches through which these questions can be answered. Nevertheless, it could be improved in its technical soundness and presentation details.

Major comments:

1. In the abstract, one of the questions is formulated as "Who were the central actors in the diffusion of these (vaccine-critical) contents?". It doesn't seem that this question can be fully answered by the corresponding conclusion "the largest right-wing community has typical echo-chamber behavior... The smaller left-wing community is less permeable ..., but has a large capacity to spread it once adopted." For example, are these two communities the central actors? What about the rest of the communities identified? Are they less central? Why? To ensure that the question matches with conclusions, I suggest that this research question be re-formulated.

2. In "Results: The mesoscale structure of the information flow", it is not clear how the two metrics -- the escape probability and the average visit probability, are calculated. This problem is partially due to limited details in the description of behaviors of the random-walker in the Method section. Different definitions could lead to very different interpretations of the results.

a. Especially, the definition of average visit probability -- "the probability (for a random walker) to visit a node of a given community" could have various explanations. For example, do we assume here the random-walker has an equal probability to start from any nodes in the network and take only one step? Or, we let the random-walker randomly walk for a large number of steps (so that the position of the random walker follows a stationary distribution) and aggregate the result from this simulation?

b. The escape probability also suffers (but potentially less) from the same issue. We probably know that the random walker is located in a community. However, do we assume the random walker has an equal chance to start from any nodes in the community? Or node with a higher out-degree in a community has a higher chance to be a starting point?

Minor comments:

1. In "Methods: Measuring the engagement dynamics", Nt is defined as the total number of active users. What's the definition of an active user?

2. In the same section, are there supportive arguments for the specific selection of the engagement window to be 3 days? Would the result significantly change if we slightly vary this parameter?

3. In Fig. 3, the y-axis label on the left doesn't make much sense. I suppose that both of blue and orange curves correspond to the number of users engaging with vaccine-critical content and news media, correspondingly. Given that the orange curve has the label "Media", the blue curve should have the label "Vaccine-critical" instead of "Engaged".

4. In Fig. 6 & 7, what does the size of each circle refers to? size of each community? Please put this information in the caption of the figure to ensure clarity.

Reviewer #3: The manuscript is partly technically sound. The selected data collection mechanism and analysis methods are suitable for addressing the research questions. Some parts in the Data and methods and Results sections need more detailed explanation:

- The APIs used in data collection are not mentioned

- There are not details on what the dataset contains (e.g. tweet ids, tweet text, number of retweets, user ids, etc.)

- The words used for the queries, how you decided to use those and which are the words?

- The qualitative analysis of the users profiles to distinguish to left or right-partite.

- There is no information within the manuscript about wether the dataset, scripts to collect the data, scripts for analysis are available for the replication of the results or future works.

The findings of this work are well supported by the data both from within the text and the figures.

The paper structure, writing style, and language is appropriate for a research manuscript.

Comments to the authors:

1. explain the selection of 3 days time window for τ, in Measuring the engagement dynamics section.

2. add the references for "compartmental models" and SIS model in epidemiology

3. define how they can say that an engaged user looses interest in β calculation

4. explain more the community detection method in the section "Community detection".

5. There is a typo in Results section, first sentence, there is the word "weather" instead of "wether"

6. PLOS authors have the option to publish the peer review history of their article (what does this mean?). If published, this will include your full peer review and any attached files.

Reviewer #1: No

Reviewer #2: No

Reviewer #3: No

---

## [Author Response · Author response to Decision Letter 0]

13 Jun 2022

See attached file.

Response to Reviewers’ comments on:

‘Assessing the influence of French vaccine critics during the two first

years of the COVID-19 pandemic’

by M. Faccin, F. Gargiulo, L. Atlani-Duault and JK Ward

June 3, 2022

We thank the editor and the reviewers for the careful evaluation of this manuscript. In particular,

we noticed the recognition of the manuscript value and soundness, despite the lack of some

important details. We spent a lot of effort in clarifying the more obscure passages suggested by

the reviewers and extending deficient parts. Specifically we extended and clarified the Methods

and Data section, with attention to explicitly reports all steps.

We considered all relevant reviewer comments and replied to each of them in the following. We

attach a document highlighting the differences from the previous submission (as compiled by

latexdiff).

on behalf of the co-authors

Mauro Faccin

Reviewer 1

Comment 1.1.

In this paper, the authors analyze the debates over the COVID-19 vaccine on the French-speaking

segment of Twitter. First, they study the evolution of vaccine-related debates. They show that specific events increase the reach of the vaccine-critical activists on social media, but this information

flow is relatively limited compared to mainstream media. Second, they analyze the community

structure of discussions and examine how information flow between communities. Authors conclude that the largest far-right community is the echo chamber of conspiracy theorists. In contrast,

a smaller community that consists of far-left actors is more capable of communicating vaccinecritical content to a broader public.

They assess the evolution of user engagement using a model similar to the SIS network epidemic

model. Here, users that share vaccine-critical information are analogous to infectious individuals

in the SIS model. Furthermore, their community analysis relies on hypergraphs derived from

retweet data. Hypergraph helps differentiate a user who is retweeted N times for a single tweet

and a user whose N tweets are individually retweeted one single time, which is crucial for capturing

the dynamics of the information flow.

In conclusion, I think this paper presents a rigorous analysis of vaccine hesitancy on Twitter during

the COVID-19 pandemic. I think the methodology is accurate, and the results are significant.

Response 1.1.

We thank the reviewer that sees an important and rigorous contribution to the literature in this

manuscript.

Comment 1.2.

(1) To the best of my understanding, "vaccine-critical URLs" refer to websites other than mainstream media. For example, websites of prominent actors in vaccine controversies. It may be

helpful for the reader if this is reminded in the data subsection.

Response 1.2.

As suggested by the reviewer we clarified the content of the URLs of interest in the Data section.

Comment 1.3.

(2) "we searched our database for those URLs" is the database refer to dataset DataCovVac?

Response 1.3.

As this passage was unclear, we clarified which database we refer (indeed the DataCovVac database)

in the main text.

Comment 1.4.

(3) I could not fully follow how the co-occurrence network is used to label URLs automatically? Did

you also propagate the labels of media URLs to closest neighbors? It appears it is performed only

for 285 vaccine-critical URLs. If this is the case, it is not clear how 382 media URLs were obtained

from the initial 50 URLs.

I think that a figure (e.g., a flowchart) that explains the data collection process might be helpful.

However, this is not a necessity.

Response 1.4.

The passage describing the database construction has been clarified and corrected in the main

text. We started from an initial seed of 285 + 50 URLs and found its dilation in the co-shared

network of URLs. From this set, we visited each website and manually classified it to either

vaccine-critical or news media or others (which we discarted).

Finally we published the code online at github.com/maurofaccin/DataCovVac

Comment 1.5.

I think the authors could mention the motivation behind using hypergraph instead of a standard

retweet network earlier in the method section. I think it is a crucial choice, and the reason is

explained in the middle of the results section.

Response 1.5.

As suggested by the reviewer we added in the method section a paragraph on the motivations

of hyper-graph choice, which depends on the lack of knowledge of the exact retweet cascade

structure and on the ability to distinguish users with many less retweeted tweets from users with

few highly retweeted tweets.

Comment 1.6.

Sometimes the COVID-19 outbreak is mentioned as an epidemic, while it is sometimes referred to

as a pandemic. Is there a nuance based on the context, or are the words "epidemic" and "pandemic" used interchangeably? For example, the title says "COVID-19 pandemic" and the abstract

says "COVID-19 epidemic" this might confuse the reader.

Response 1.6.

To avoid any ambiguity we normalized the use of “pandemic”.

Reviewer 2

Comment 2.1.

In this study, the authors investigated the influences and spreading of vaccine-critical content

on social media. They focused on Twitter data and applied network analysis tools to answer two

questions: (1) Did vaccine-critical contents exhibit a "rise" during the COVID breakout? (2) What

are the roles that different communities (groups of closely-connected Twitter users) play in the

flow of vaccine-critical information.

Generally, the draft is clear about the questions and the general approaches through which these

questions can be answered. Nevertheless, it could be improved in its technical soundness and

presentation details.

Response 2.1.

We thank the reviewer for spotting the manuscript potential and for helping us to improve its

soundness and rigorousness.

Comment 2.2.

1. In the abstract, one of the questions is formulated as "Who were the central actors in the

diffusion of these (vaccine-critical) contents?". It doesn't seem that this question can be fully

answered by the corresponding conclusion "the largest right-wing community has typical echochamber behavior... The smaller left-wing community is less permeable ..., but has a large capacity

to spread it once adopted." For example, are these two communities the central actors? What

about the rest of the communities identified? Are they less central? Why? To ensure that the

question matches with conclusions, I suggest that this research question be re-formulated.

Response 2.2.

We thank the reviewer for rising this source of confusion. Our aim was to study and analyse the

influence of those communities in the diffusion of vaccine-critical content, without referring to

any notion of centrality (measure of betweenness or random walk centrality for example.) We

amended all parts of the text referring to this concepts. On the other hand, those two communities are the main actors in the diffusion of such contents, in fact they have the highest probability

of being traversed by a tweet (visiting probability) while displaying, in the left-wing case, a non

negligible ability to reach other communities (escape probability).

Comment 2.3.

2. In "Results: The mesoscale structure of the information flow", it is not clear how the two metrics

-- the escape probability and the average visit probability, are calculated. This problem is partially

due to limited details in the description of behaviors of the random-walker in the Method section.

Different definitions could lead to very different interpretations of the results. a. Especially, the

definition of average visit probability -- "the probability (for a random walker) to visit a node of a

given community" could have various explanations. For example, do we assume here the randomwalker has an equal probability to start from any nodes in the network and take only one step? Or,

we let the random-walker randomly walk for a large number of steps (so that the position of the

random walker follows a stationary distribution) and aggregate the result from this simulation? b.

The escape probability also suffers (but potentially less) from the same issue. We probably know

that the random walker is located in a community. However, do we assume the random walker

has an equal chance to start from any nodes in the community? Or node with a higher out-degree

in a community has a higher chance to be a starting point?

Response 2.3.

The characterization of the random walk considered in this manuscript is described in the Methods, in particular in the Dynamics and hyper-graphs subsection and is as follows. At a given

time, the random walk resides on a node (a user of Twitter). From this node the random walk

choose with an even probability one of the hyper-edges incident to the node by its tail (any of

the tweets produced by that user). Once selected the hyper-edge, the random walker select one

of the head nodes with even probability. This let us define the transition probability p( j|i ), and

the visiting probability of each node p(i ) is computed at the steady state by an iterative algorithm. The community transition matrix is:

p(C 0 |C ) =

∑i∈C0 ,j∈C p( j|i ) p(i )

∑i ∈C p (i )

(1)

From here one can compute the visiting probability of community C as ∑i∈C p(i ) and its per-user

average value as ∑i∈C p(i )/|C |. The escape probability from community C is defined as

p(C |C ) = ∑ p(C 0 |C )

(2)

C 0 6=C

We have clarified the Methods section to contain these definitions.

Comment 2.4.

1. In "Methods: Measuring the engagement dynamics", Nt is defined as the total number of active

users. What's the definition of an active user?

Response 2.4.

We clarity in the text that in the context of engagement analysis, active users are those that

tweet or retweet on that day.

Comment 2.5.

2. In the same section, are there supportive arguments for the specific selection of the engagement window to be 3 days? Would the result significantly change if we slightly vary this parameter?

Response 2.5.

We thanks the reviewer for rising this question. The results are robust as long as the time window

is kept small, less than a week. On the other hand, for the purpose of computing the engagement

of users in a social network whose dynamics are fast-paced, a short time-window would better

capture how it changes. We mentioned the robustness of the method on the Methods section.

Comment 2.6.

3. In Fig. 3, the y-axis label on the left doesn't make much sense. I suppose that both of blue

and orange curves correspond to the number of users engaging with vaccine-critical content and

news media, correspondingly. Given that the orange curve has the label "Media", the blue curve

should have the label "Vaccine-critical" instead of "Engaged".

Response 2.6.

We thank the reviewer for spotting this labeling inconsistency which has been amended in the

new version of this manuscript.

Comment 2.7.

4. In Fig. 6 & 7, what does the size of each circle refers to? size of each community? Please put

this information in the caption of the figure to ensure clarity.

Response 2.7.

We have fixed the lack of information spotted by the reviewer.

Reviewer 3

Comment 3.1.

The manuscript is partly technically sound. The selected data collection mechanism and analysis

methods are suitable for addressing the research questions. Some parts in the Data and methods

and Results sections need more detailed explanation:

• The APIs used in data collection are not mentioned

• There are not details on what the dataset contains (e.g. tweet ids, tweet text, number of

retweets, user ids, etc.)

• The words used for the queries, how you decided to use those and which are the words?

• The qualitative analysis of the users profiles to distinguish to left or right-partite.

• There is no information within the manuscript about wether the dataset, scripts to collect

the data, scripts for analysis are available for the replication of the results or future works.

Response 3.1.

We thank the reviewer for recognizing the value of the manuscript.

Following the reviewer suggestion we clarified and added details to various passages of the

article.

• We already mentioned that we used both search and stream APIs; we additionally referenced other papers were an in-depth description of the dataset extraction is discussed and

upload the list of used keywords to a public repository at github.com/maurofaccin/DataCovVac.

• We feel that the list of metadata contained in the extracted dataset would distract the

reader from the main message of the article, we choose not to report this.

• For what concern the keywords used in the dataset extraction, the complete list has been

uploaded to a public repository and referenced in the main text. Those keywords were

selected based on previous analysis which has been explicitly referenced in the main text.

• We amended the main text shortly mentioning the availability of the dataset and the software used to extract it. The full dataset cannot be shared due to current Twitter policies,

but a list of tweet IDs can be provided upon request.

Comment 3.2.

The findings of this work are well supported by the data both from within the text and the figures.

The paper structure, writing style, and language is appropriate for a research manuscript.

Response 3.2.

We thank the reviewer for recognizing the soundness of the manuscript.

Comment 3.3.

1. explain the selection of 3 days time window for τ, in Measuring the engagement dynamics

section.

Response 3.3.

We added a deeper discussion on the choice. Particularly we stress the robustness of the analysis

on the modification of the temporal window.

Comment 3.4.

2. add the references for "compartmental models" and SIS model in epidemiology

Response 3.4.

We added a reference for the SIS model and related theoretical results.

Comment 3.5.

3. define how they can say that an engaged user looses interest in β calculation

Response 3.5.

We clarified how the calculation works.

In particular, at any day t we can compute the following quantities from the dataset:

Et the number of users that shared a link from the set of URLs within the time frame (t − τ, t];

Nt the number of users that tweeted anything within the time frame (t − τ, t];

dEt+ the number of users that were engaged at time t − 1 but uncommitted at time t;

dEt− the number of users that were uncommitted at time t − 1 but engaged at time t;

from the above and Eq. 1 and 2 in the manuscript, one can estimate the engagement and disengagement rates αt and β t at any time t and consequently the reproduction number Rt from Eq.

4.

Comment 3.6.

4. explain more the community detection method in the section "Community detection".

Response 3.6.

We clarified how the communities were computed, focusing in how to the stability algorithm was

used in our case.

Comment 3.7.

5. There is a typo in Results section, first sentence, there is the word "weather" instead of "wether"

Response 3.7.

This typo has been fixed.

---

## [Editor Report · Decision Letter 1]

27 Jun 2022

Assessing the influence of French vaccine critics during the two first years of the COVID-19 pandemic

PONE-D-22-05617R1

Dear Dr. Faccin,

We’re pleased to inform you that your manuscript has been judged scientifically suitable for publication and will be formally accepted for publication once it meets all outstanding technical requirements.

Kind regards,

Constantine Dovrolis

Academic Editor

PLOS ONE
---

## [Editor Report · Acceptance letter]

27 Jul 2022

PONE-D-22-05617R1 

Assessing the influence of French vaccine critics during the two first years of the COVID-19 pandemic 

Dear Dr. Faccin:

I'm pleased to inform you that your manuscript has been deemed suitable for publication in PLOS ONE. Congratulations! Your manuscript is now with our production department. 

Kind regards, 

on behalf of

Dr. Constantine Dovrolis 

Academic Editor

PLOS ONE